# Identification of Missense Variants Affecting Carcass Traits for Hanwoo Precision Breeding

**DOI:** 10.3390/genes14101839

**Published:** 2023-09-22

**Authors:** Dong Jae Lee, Yoonsik Kim, Phuong Thanh N. Dinh, Yoonji Chung, Dooho Lee, Yeongkuk Kim, Soo Hyun Lee, Inchul Choi, Seung Hwan Lee

**Affiliations:** 1Division of Animal & Dairy Science, Chungnam National University, Daejeon 34134, Republic of Korea; leedj636@gmail.com (D.J.L.); yoongi1565@naver.com (Y.C.); stonecold@daum.net (D.L.); lhyungm@gmail.com (S.H.L.); 2Department of Bio-AI Convergence, Chungnam National University, Daejeon 34134, Republic of Korea; watogo@hanmail.net (Y.K.); dnphuongthanh1511@gmail.com (P.T.N.D.); 3Quantomic Research & Solution, Daejeon 34134, Republic of Korea; lgs10kr@naver.com

**Keywords:** carcass traits, exon-specific association study, missense variant, protein structure prediction

## Abstract

This study aimed to identify causal variants associated with important carcass traits such as weight and meat quality in Hanwoo cattle. We analyzed missense mutations extracted from imputed sequence data (*ARS-UCD1.2*) and performed an exon-specific association test on the carcass traits of 16,970 commercial Hanwoo. We found 33, 2, 1, and 3 significant SNPs associated with carcass weight (CW), backfat thickness (BFT), eye muscle area (EMA), and marbling score (MS), respectively. In CW and EMA, the most significant missense SNP was identified at 19,524,263 on BTA14 and involved the *PRKDC*. A missense SNP in the *ZFAND2B*, located at 107,160,304 on BTA2 was identified as being involved in BFT. For MS, missense SNP in the *ACVR2B* gene, located at 11,849,704 in BTA22 was identified as the most significant marker. The contribution of the most significant missense SNPs to genetic variance was confirmed to be 8.47%, 2.08%, 1.73%, and 1.19% in CW, BFT, EMA, and MS, respectively. We generated favorable and unfavorable haplotype combinations based on the significant SNPs for CW. Significant differences in GEBV (Genomic Estimated Breeding Values) were observed between groups with each favorable and unfavorable haplotype combination. In particular, the missense SNPs in *PRKDC*, *MRPL9*, and *ANKFN1* appear to significantly affect the protein’s function and structure, making them strong candidates as causal mutations. These missense SNPs have the potential to serve as valuable markers for improving carcass traits in Hanwoo commercial farms.

## 1. Introduction

Hanwoo cattle, endemic to the Republic of Korea, has been residing on the Korean Peninsula for over 2000 years. The refinement of Hanwoo is achieved by intensive selection focusing on economic traits, such as meat quantity and quality [1]. Following the human genome project, advances in whole-genome sequencing of livestock have led to the development of SNP arrays, which are further expedited by genome-wide panels of SNPs to enable genomic prediction in livestock. The number of genetic variations (SNPs, insertions, deletions, and structural variations) available for genomic prediction and the number of whole-genome sequenced individuals is increasing [2]. The 1000 Genomes Project consortium reported 38 million SNPs with 1.4 million short insertions and deletions in humans [3]. The 1000 Bull Genomes Project identified 28.3 million variants including insertions, deletions, and SNPs in *Bos taurus* [4].

Genome-wide association studies (GWAS) have become an essential tool for identifying genetic markers associated with complex quantitative traits in animal genetics. These studies have successfully identified a large number of trait-associated single nucleotide polymorphisms (SNPs) [5,6,7]; however, the polygenic structures are very difficult to characterize. The missing heritability problem is a significant challenge in GWAS studies, as the genetic variation for polygenic traits has a minor effect, accounting for only a small fraction of heritability [8,9]. In addition, complex traits are affected by genetic interactions and environments, and numerous SNPs affect target traits due to a high level of linkage disequilibrium (LD) between genomic variations, making it difficult to specify the exact cause of a genetic variation [10].

In order to overcome the challenges of identifying genetic variants associated with complex traits in GWAS, many studies have used biological information to annotate classes of mutations and search for trait-associated mutations in specific regions. In human studies, protein-coding and conserved annotations have been used to identify trait-related variants [11,12,13]. We here focused on missense (non-synonymous) variants that are highly associated with trait-associated SNPs [14,15] because they can impair protein stability or cause structural changes that lead to functional alterations [16]. It has been shown that the effects of amino acid allelic variants on protein structure and function can be predicted by analysis of multiple sequence alignments [17,18]. Multiple sequence alignments of homologous sequences identify which positions have been conserved over evolutionary time, and these positions are inferred to be important for function [19]. At CASP14 in 2020, AlphaFold2 (AF2) demonstrated that high-precision protein structure prediction with high accuracy on an unprecedented scale is not only possible but has already been achieved [20,21]. In 2021, DeepMind and EMBL-EBI developed the AlphaFold protein structure database, a data resource that presents protein structure prediction results and makes them easily accessible to the community, making a huge impact on the life sciences field [22,23].

In fact, SNP prioritization approaches using missense SNPs have been developed to predict disease associated bio-markers [24]. In the context of livestock breeding programs, missense variants in the protein-coding region account for a significant proportion of the genetic variance in dairy and beef cattle [2]. This makes the exon region a promising target for identifying genetic variants associated with complex traits in livestock [25]. In this study, we aim to identify causal mutations that affect carcass traits in Hanwoo cattle by conducting an association analysis that takes into account the functional aspects of SNPs, particularly missense variants. In addition, this study is intended to help understand the effect of the missense mutation on the protein structure.

## 2. Materials and Methods

### 2.1. Animals, SNP Genotyping, and Quality Control

This study used 16,970 commercial Hanwoo with an average age of 30.2 ± 1.9 months that were born between 2006 and 2016. These animals were slaughtered at a commercial abattoir between 2013 and 2019, producing phenotypic data (carcass traits). The genomic DNA was derived from *longissimus thoracis* muscle samples using a DNeasy Blood and Tissue Kit (Qiagen, Valencia, CA, USA), and its concentration and purity were assessed using a NanoDrop 1000 (Thermo Fisher Scientific, Wilmington, DE, USA). DNA samples were genotyped using the Bovine SNP50 BeadChips (Illumina Inc., San Diego, CA, USA). Quality control was performed using PLINK1.9 software [26] based on the following SNP exclusion criteria: call rate < 0.90; minor allele frequency < 0.01; and *p*-value for Hardy–Weinberg equilibrium test <10^−4^.

### 2.2. Imputation, Lift-Over, and Annotation

Imputation was conducted in a two-step process using minimac3 [27]. Initially, the 50 K genotype of 16,970 Hanwoo was uplifted to a high-density level using the Illumina Bovine HD BeadChip (777K) from 1295 Hanwoo samples. Subsequently, it was further imputed to the sequence level utilizing whole genome sequence data from 203 Hanwoo. SNPs with square correlation (r^2^) < 0.6 were excluded in the imputation steps, and SNPs on the autosomes were selected from the analysis. The imputed sequence-level data was annotated using the SnpEff program [28]. Genomic coordinates were converted from *UMD 3.1.1* to *ARS-UCD1.2*.

### 2.3. Variance Component Estimation of Genome Regions

We estimated the variance explained by genome regions via restricted maximum likelihood (REML) analysis using GCTA software [29]. The statistical model was
(1)y=Xβ+g+e
where y is a vector of the carcass traits, X is an incidence matrix, β is a vector of fixed effects, g is a vector of genetic effects of genome regions with var(g) = $A_{g}\sigma_{G}^{2}$, and $A_{g}$ is the genomic relationship matrix (GRM) for all genome regions. The proportion of variance explained by genome regions was defined as $h_{G}^{2}$ = $\sigma_{G}^{2}/\sigma_{P}^{2}$.

### 2.4. Exon-Specific Association Test (ESAS)

The genome region-specific genomic data was obtained by extracting only non-synonymous (missense) variants from the imputed sequence-level data. The phenotypic data were pre-adjusted for fixed effects, including birth year, season of birth, growing regions, and age at slaughter, utilizing a linear mixed model in R software 3.3.1 (R Foundation for Statistical Computing, Vienna, Austria). The adjusted phenotypes and SNP information were subsequently submitted for GWAS using a mixed linear model. It can be written as:(2)y=a+bx+g+e


y: phenotype; a: the mean term; b: additive effect (fixed effect) of the candidate SNP; x: SNP genotype indicator variable coded as 0, 1, or 2, which are the number of minor alleles of the genotype; g: polygenic effect (random effect); e: residual.

The significance test in the exon-specific association test was performed using the Bonferroni correction method (0.05/number of SNPs). The percentage of genetic contribution (%Vg) accounted for by each SNP was calculated using the formula
(3)$$\%V_{g}=100\;\times \;\frac{2\mathrm{pq}\beta^{2}}{\sigma_{A}^{2}}$$

The explanation power of a candidate SNP in phenotypic variance (total variance) can be calculated as follows [30]:(4)$$\frac{\beta^{2}\mathrm{Var}(X)}{\mathrm{Var}(Y)}=\frac{\beta^{2}\mathrm{Var}(X)}{\beta^{2}\mathrm{Var}\left(X\right)+\sigma^{2}}$$

β: effect size of genetic variant (X); β^2^Var(X): variance explained by the genetic variant X; Var(Y): phenotypic variance; σ^2^: variance explained by other genetic variants. It can be estimated as:(5)$$\frac{2{\hat{\beta }}^{2}M(1-M)}{2{\hat{\beta }}^{2}M\left(1-M\right)+(se(\hat{\beta }){)}^{2}2NM(1-M)}=\frac{{2pq\hat{\beta }}^{2}}{Var(Y)}$$

$\hat{\beta }$: effect size estimate; M: minor allele frequency; N: sample size; se($\hat{\beta }$): standard error of effect size for the genetic variant X; q: major allele frequency.

### 2.5. Favorable and Unfavorable Haplotypes

The haplotype phase was inferred from the significant SNPs of each chromosome for carcass weight using PHASE v2.1.1 [31,32], using 100 iterations, two thinning intervals per iteration, and 10 burn-in iterations. By employing the effects and allelic information of significant SNPs associated with body weight, we delineated favorable and unfavorable homozygous haplotypes (HH) within the reconstructed haplotypes of each chromosome.

### 2.6. Genomic Prediction

The GEBVs of all genotyped individuals were predicted with the genomic best linear unbiased prediction (GBLUP) model using the BLUPF90 programs [33]. GEBVs were calculated based on the following equation:(6)y=1μ+Zγ+g+e
where y is the vector of the observed phenotype; 1 is a vector with N ones; μ is the intercept; Z is the incidence matrix for the fixed effects; γ is the vector of fixed effects, which included birth year, birth month, slaughter year, slaughter month, slaughter place, and age as covariates for traits; g is vector of the additive genetic effects for individuals with var$\left(g_{i}\right)=G_{i}\sigma_{a}^{2}$; and e is the vector of residual effects. G is the genomic relationship matrix calculated as:(7)$$G=\frac{MM^{\prime }}{2\sum p_{j}(1-p_{j})}$$
where M is a matrix of centered genotypes and $p_{j}$ is the allele frequency for marker j [34].

### 2.7. Prediction of Damaging Causal Mutations and Structure

We used PolyPhen-2 v2 [17] to predict the impact of the candidate causal genetic mutation (missense mutation) on the protein structure and function. Furthermore, for those mutations with significant impact, we utilized AlphaFold2 [35] to predict the 3D protein structure based on the presence or absence of the mutation.

## 3. Results

### 3.1. Genome Partitioning of Genetic Variation

In this study, we obtained 693,496 SNPs (90%) out of 777K SNPs and 25,676,502 SNPs (100%) of whole-genome sequence data. Applying the R-square criterion, imputed sequence-level data consisting of 10,443,247 SNPs were generated, and the majority of mutations were found in intron regions at 53.16%, followed by intergenic regions at 37.09%. Exons accounted for only 0.83% of the entire genome, a relatively small proportion (Figure 1A).

In terms of heritability by genome region, the intergenic region had the highest values for carcass weight (CW), backfat thickness (BFT), and eye muscle area (EMA) at 0.22, 0.16, and 0.14, respectively, while the highest value for marbling score (MS) was observed in the intron region at 0.18. Exon regions had lower heritability values of 0.06, 0.03, 0.03, and 0.01 for CW, BFT, EMA, and MS, respectively (Figure 1B).

Currently, the common chips universally used in Hanwoo are Illumina BovineSNP50 v2, Illumina BovineSNP50 v3, and the customized Hanwoo 50K chip. The number of SNPs incorporated in each chip is 54,609 for Illumina v2, 53,218 for Illumina v3, and 53,866 for the Hanwoo custom chip (Figure 1C). As a result of checking whether 28,144 missense variants in the exon region were inherent in these, 135 missense SNPs out of 28,144 were found in common, and 27,996 missense SNPs were not included in the currently produced chip.

### 3.2. Identification of Candidate Causal Variants

Based on the quantile–quantile (QQ) plot for each carcass trait, the genomic inflation factor (lambda value) was observed to range from 0.82 to 0.94 (Appendix A). Only the missense variants that passed the quality control criteria were used for the exon-specific association test, and the results are presented in Figure 2. The results showed 33, 2, 1, and 3 significant SNPs associated with CW, BFT, EMA, and MS, respectively. Information on significant SNPs for each trait is shown in Table 1, and more details are shown in Appendix A. The most significant SNP for CW was the marker rs449968016 located at position 19,524,263 bp in chromosome BTA14, within the protein kinase, DNA-activated, catalytic subunit (*PRKDC*) gene. The analysis revealed a significant association between the EMA and the same SNP (rs449968016). The most significant marker (rs109446852) for BFT was identified at position 107,160,304 bp in chromosome BTA2, within an exon of the zinc finger AN1-type containing the 2B (*ZFAND2B*) gene. Moreover, the most significant marker (rs799031002) for MS was located in the activin A receptor type 2B (*ACVR2B*) gene at position 11,849,704 bp on chromosome BTA22. Regarding CW, significant SNPs were identified in each of chromosomes BTA6 (rs797342426), BTA14 (rs471616366), and BTA14 (rs380389290). These SNPs correspond to hypothetical genes that have not yet been identified in cattle.

### 3.3. Explanatory Power of Candidate SNPs

To estimate the effect of missense variants, we analyzed the extent to which each SNP contributes to the genetic variation and phenotypic variance. The results are shown in Appendix A. For CW, 28,093 SNPs (99.82%) out of a total of 28,144 missense mutations did not explain 1% of the genetic variance, and only 51 SNPs (0.18%) were found to explain 1.03~8.47%. Additionally, 28,130 SNPs (99.95%) explained less than 0.01% of the phenotypic variance, and only 14 SNPs were found to contribute 0.01~0.03% phenotypic variance. BFT showed that only 7 SNPs (0.02%) explained 1.03~2.07% of the genetic variance, and only 1 SNP explained more than 0.01% of the phenotypic variance. In the EMA, only 3 SNPs (0.01%) explained 1.01 to 1.73% of the genetic variance, and none explained more than 0.01% of the phenotypic variance. In MS, only 2 (0.01%) SNPs explained 1.10 to 1.19% genetic variance, and no SNPs were identified that explained phenotypic variance greater than 0.01%. The contributions to genetic and phenotypic variance of the most significant SNPs for carcass traits on each chromosome are presented in Table 2.

The SNP with the highest explained genetic and phenotypic variation for CW was rs449968016, with explanatory power of 8.470% and 0.026%, respectively. This result was consistent with the most significant SNP identified in the exon-specific association test. Notably, this marker was the only one present in the Hanwoo custom chip of all SNPs that appeared significant for carcass traits. For BFT, one SNP (rs109446852) explained the highest genetic variance at 2.075% and phenotypic variance at 0.006%, consistent with the most significant SNP identified in this study. In EMA, the SNP rs449968016 equivalent to carcass weight explained the highest genetic variation at 1.728% and was found to contribute 0.005% to phenotypic variance. For MS, the SNP rs799031002, the most significant in the exon-specific GWAS, was found to explain the highest genetic variation at 1.187% and accounted for 0.004% of the phenotypic variance.

### 3.4. Favorable and Unfavorable Homozygous Haplotypes

Haplotype estimation for each chromosome using the 33 SNPs significantly present in carcass weight identified 16, 24, and 1103 haplotypes in BTA4, BTA6, and BTA14, respectively. Favorable homozygous haplotypes and unfavorable homozygous haplotypes were selected based on the effects of significant SNPs on carcass weight. In the BTA4 region of all individuals, 78 individuals had a favorable homozygous haplotype (AACCCC-GG), and 5518 individuals had an unfavorable homozygous haplotype (GGTTTTAA). For the BTA6 region, there were no individuals with a favorable homozygous haplotype (GGTTCCCCGGTTGG), while 916 individuals had an unfavorable homozygous haplotype (AACCTTTTTTTCCAA). Similarly, in the BTA14 region, there were no individuals with a favorable homozygous haplotype (AAAAGGTTAATTTTTTCCGGTTTTTAATTC-CGGCCTTTTAAAAAA), and only 18 individuals had an unfavorable homozygous haplotype (TTGGAACCGGCCCCAATTTTCCCCGGGGGTTAAAACCCCGGGGGG). Since no individuals with favorable homozygous haplotypes in BTA6 and BTA14 were identified, to pinpoint the optimal beneficial haplotype combination, if at least five individuals appeared that were as similar as possible to the favorable homozygous haplotype, the corresponding haplotype was selected as a favorable haplotype. It was confirmed that there were 25 animals in BTA6 and 6 individuals in BTA14. In order to distinguish additive genetic differences between individuals, with respect to carcass weight, with estimated combinations of favorable and unfavorable haplotypes, the genomic breeding values of individuals with favorable and unfavorable haplotypes for chromosomes BTA4, BTA6, and BTA14 from the distribution of genetic breeding values of 16,970 Hanwoo were determined. The genetic differences between groups with favorable and unfavorable allele combinations were analyzed for statistical significance using independent t-tests, and the results are presented in Figure 3.

In BTA4, the average GEBV and standard deviation for individuals with favorable and unfavorable homozygous haplotypes were 24.62 ± 23.68 and −0.77 ± 26.67, respectively, and a significant difference between the two groups was confirmed. For the BTA6, individuals with favorable and unfavorable haplotype combinations showed an average GEBV and standard deviation of 47.07 ± 33.39 and −2.98 ± 25.26, respectively, with a significant difference observed between the two groups. Similarly, in BTA14, the average GEBV and standard deviation for those with favorable and unfavorable haplotype combinations were 41.46 ± 27.33 and 0.63 ± 20.29, respectively, presenting a significant difference between the groups.

### 3.5. Effects of Causal Variants on Protein Structure

PolyPhen-2 predicts the effects of amino acid substitutions on proteins by considering their sequence, phylogenetic, and structural information. We predicted the impact on proteins when mutations targeted the missense SNPs most significantly associated with the carcass traits and a substantially influenced genetic and phenotypic variance. The results are presented in Table 3.

The protein kinase, DNA-activated, catalytic subunit (*PRKDC*) missense variant (c.2683G>A; p.Ala895Thr) exhibited a damaging score of 0.988, indicating that it is deleterious and has a significant impact on the protein’s function and structure. Additionally, the missense variant (c.676A>G; p.Thr226Ala) of mitochondrial ribosomal protein L9 (*MRPL9*) and missense variant (c.520C>G; p.Leu174Val) of the ankyrin repeat and fibronectin type III domain containing 1 (*ANKFN1*) showed damaging scores of 0.988 and 0.961, respectively, confirming that they have a detrimental impact on the protein. In contrast to previous results, the cordon-bleu WH2 repeat protein (*COBL*) missense variant (c.1979G>A; Arg660Gln) and the zinc finger AN1-type containing the 2B (*ZFAND2B*) missense variant (c.350T>G; Leu117Arg) are tolerated, indicating they do not affect protein function or structure. The impact on protein function due to the amino acid substitution of a ligand-dependent nuclear receptor corepressor like (*LCORL*) missense variant (c.1453G>A; p.Glu485Lys) has not been estimated as there is no reported research data on it.

We predicted the 3D structure of the PRKDC, MRPL9, and ANKFN1 proteins, which are anticipated to impact the function or structure of the protein (Figure 4). The multiple sequence alignment (MSA), predicted local distance difference test (pLDDT) score per position, and predicted aligned error (PAE) score for each protein are shown in Appendix A. As a result of comparing the wild-type and mutant protein structures of each protein, it was observed that the protein had a different 3D structure, implying that changes in amino acids due to missense variants influence the protein.

## 4. Discussion

The effects of each SNP on the carcass traits of Hanwoo are minimal and exhibit a polygenic structure (Figure 1 and Figure 2). It implies a high degree of missing heritability, indicating that searching for causal variants is challenging due to their genetic complexity. This study focused on the functional effects of SNPs in an attempt to understand the genetic mechanisms underlying complex traits and prioritized the exploration of genetic markers that alter protein structures due to single changes in DNA sequences.

Upon examining previous research on QTL regions for Hanwoo carcass traits, the major QTLs for CW were identified on chromosomes 4, 6, and 14; EMA on chromosome 14; BFT on chromosome 2; and MS content on chromosomes 3, 19, and 22. These observations align with the findings of our study [7,36,37]. Interestingly, out of the 33 significant missense mutations we identified for CW, the same SNPs (rs449968016, rs109953090, rs381116984) appeared in the *PRKDC*, DnaJ heat shock protein family (*Hsp40*) member C5 β (*DNAJC5B*) and corticotropin-releasing hormone (*CRH*) genes located on BTA14 [38]. Furthermore, the same missense mutation (rs449968016) in *PRKDC* was detected in EMA, consistent with our study.

For CW, the cordon-bleu WH2 repeat protein (*COBL*), most significantly manifested on chromosome 4, and is a member of the G/F-actin binding protein family acting as an actin nucleator with WH2 domains [39,40]. *COBL* plays a pivotal role in axis patterning during developmental processes. Utilizing repeated WH2 domains, it interacts with actin, undertaking G-actin sequestration, actin filament nucleation, filament severing, and barbed-end dynamics regulation. All of these are essential for modulating cellular morphogenesis by controlling cytoskeletal dynamics [41,42]. The significant missense mutation (rs210475204) discovered in the gene indicated no major impact on protein function or structure due to the single amino acid sequence change. This implies that the COBL protein might be influenced by the combined actions of other SNPs.

In agreement with previous studies—where specific *NCAPG-LCORL* loci that are significantly associated with feed intake for weight gain and feed intake in beef cattle [43], and body skeleton size in the Japanese Black steer and F2 bulls of a Charolais x German Holstein cross [44]—we also found an SNP c.1326T>G within the *NCAPG* (non-SMC condensin I complex, subunit G) gene associated with carcass traits. Additionally, it has been demonstrated that the *LCORL* region is related to growth traits in European and African *Bos taurus* [45], and the expression of *LCORL* has been confirmed to be associated with feed intake in bovine muscle tissue [46]. Similarly, in our study, the missense mutation in *LCORL* (c.1453G>A; p.Glu485Lys) appeared to influence approximately 4.5% of the genetic variance, representing the most substantial influence on BTA6. Unfortunately, we could not determine the impact of this specific missense mutation on the protein’s structural alteration due to the amino acid substitution.

The *PRKDC* gene is essential for embryonic development, interferon tau expression, and trophoblast development rate and has been reported to be involved in early bovine embryo development [47]. This gene encodes the catalytic subunit of the DNA-dependent protein kinase (DNA-PKcs), which is a member of the phosphatidylinositol 3-kinase-related kinase (PIKK) protein kinase family and a serine/threonine protein kinase [48]. Earlier studies [49] reported that DNA-PK phosphorylates threonines 5 and 7 of HSP90a reduces the chaperone function of clients crucial for mitochondrial biogenesis and energy metabolism, such as AMP-activated protein kinase (AMPK). When DNA-PK activity is reduced, AMPK activity increases. This observed weight gain in middle-aged mice suggests that a similar mechanism may operate in the end product, carcass weight. The *PRKDC* missense variant (p.Ala895Thr) undergoes a side-chain transition from hydrophobic to hydrophilic (uncharged) amino acids, changing the amino acid’s chemical properties. This shift, due to a single amino acid variation (SAV), results in a protein structure distinct from the norm (Figure 4A), indicating the possibility that this mutation might be a causal variant.

The *ZFAND2B* gene encodes a protein containing an AN1-type zinc finger and a ubiquitin interaction motif and regulates the insulin-like growth factor receptor signaling pathway [50]. The significant missense mutation (rs109446852) that accounts for about 2% of genetic variance results in a single amino acid substitution that does not affect protein function or structure. The impact may be attributed to other mutations with various functional effects, excluding this one.

In BTA3, the *MRPL9* missense variant (p.Thr226Ala) most significantly associated with marbling exhibited a change in protein structure due to its transition from hydrophilic (uncharged) to hydrophobic amino acids. This alteration implies that it might have different roles in interactions with other proteins and various biochemical reactions. The Mitochondrial Ribosomal Protein L9 (*MRPL9*) gene is a critical component of its gene product in cell proliferation and protein biosynthesis [51].

For BTA19, the most significant *ANKFN1* missense variant (p.Leu174Val) related to MS involves an amino acid change, but both residues fall under hydrophobic amino acids; even though their chemical properties are similar, they were found to have distinct protein structures. The *ANKFN1* gene has been implicated in determining human height [52,53]. However, there have been no reported functional studies on this gene in cattle, and it may potentially be an important candidate gene for marbling.

Activin-type II receptor B (*ACVR2B*) is a gene linked to myostatin and plays a role in its signaling and control. In cattle, genetic mutations in myostatin lead to prominent effects on skeletal muscle, manifesting as a dual muscle characteristic in certain breeds [54].

## 5. Conclusions

This study used a functional GWAS approach focused on the missense variants causing amino acid sequence changes to identify markers associated with carcass traits in Hanwoo cattle. Our results suggest that missense variants may impact carcass trait performance by altering amino acids, even though most candidate markers had little contribution to genetic variance. The haplotype combination of significant SNPs identified in this study can be used as a valuable marker for genome prediction in Korean cattle breeding programs. Significantly, the missense mutations in *PRDKC*, *MRPL9*, and *ANKFN1* account for a significant portion of the genetic and phenotypic variance in CW and MS, with the change in the single amino acid sequence having a critical impact on the protein’s function and structure, suggesting a high probability of these mutations being causal variants. These findings can provide insights into understanding the genetic factors influencing the carcass traits of Hanwoo cattle, and further functional research is necessary to fully comprehend the genetic mechanisms behind these findings.

## Figures and Tables

**Figure 1 genes-14-01839-f001:**
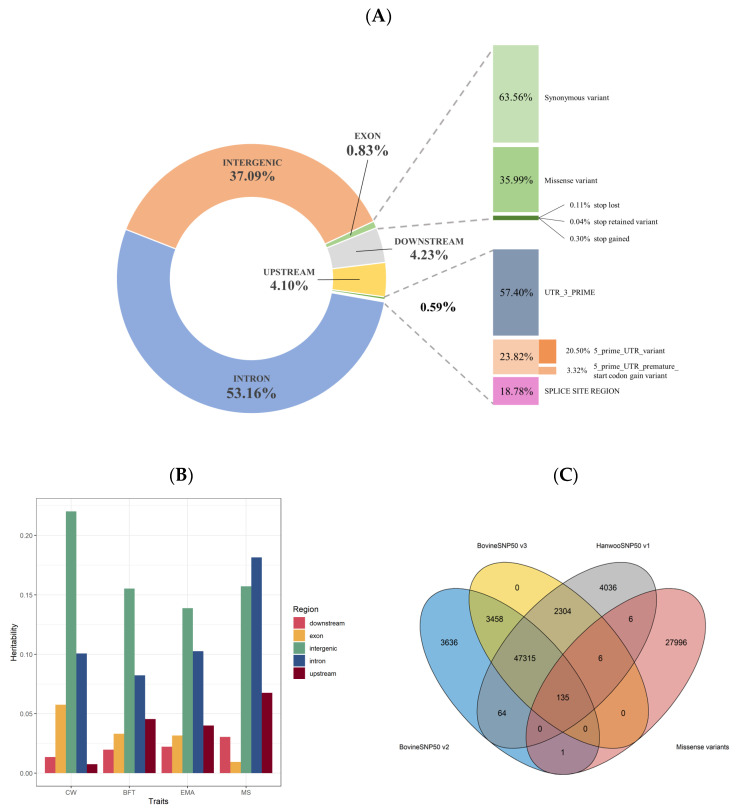
Comprehensive information on missense variants in the Hanwoo genome. (**A**) Functional effects of mutations on the sequence-level data. (**B**) The heritability of variants by genome region. (**C**) Missense mutations in Illumina Bovine SNP50 BeadChips (BovineSNP50 v2, BovineSNP50 v3) and the customized Hanwoo 50K chip.

**Figure 2 genes-14-01839-f002:**
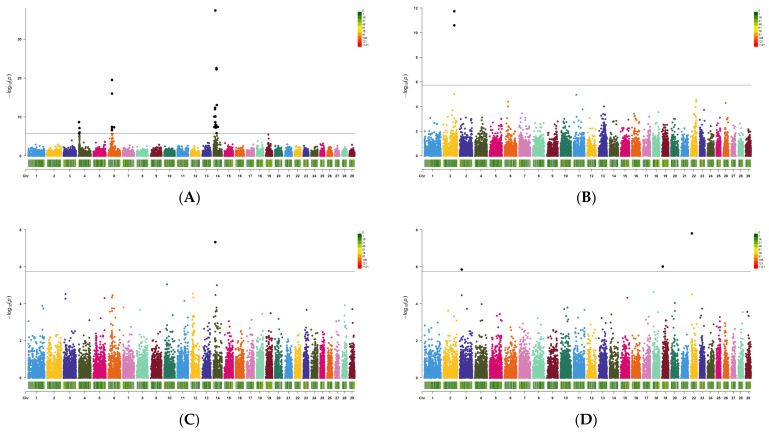
Manhattan plots of the exon-specific association study (ESAS) using missense variants for carcass traits. (**A**) CW; (**B**) BFT; (**C**) EMA; and (**D**) MS.

**Figure 3 genes-14-01839-f003:**
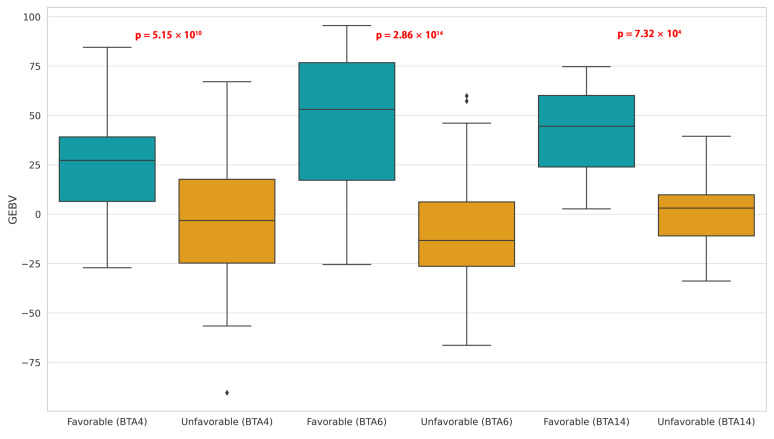
Distribution of individuals’ GEBV depending on the type of haplotype combination (favorable vs. unfavorable) in BTA4, BTA6, and BTA14.

**Figure 4 genes-14-01839-f004:**
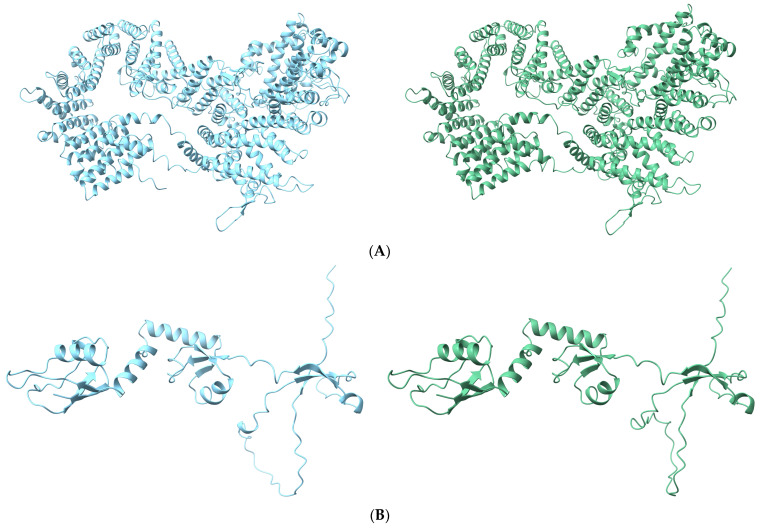
Protein structure prediction of wild-type and mutant-type for protein kinase, DNA-activated, catalytic polypeptide (PRKDC), mitochondrial ribosomal protein L9 (MRPL9), and ankyrin repeat and fibronectin type III domain containing 1 (ANKFN1) protein. (**A**) 3D protein structures of wild-type (left) and mutant-type (right) PRKDC protein using 1400 amino acids including the target sequence (p.Ala895Thr); (**B**) MRPL9 protein structure with wild-type (left) and mutant-type (right) (p.Thr226Ala); (**C**) ANKFN1 protein structure with wild-type (left) and mutant-type (right) (p.Leu174Val).

**Table 1 genes-14-01839-t001:** The candidate SNPs and genes associated with carcass traits in Hanwoo cattle.

Trait	Chr:Pos	Allele	RefSNPID	Gene	SNPeffect	SE	*p*-Value
CW	4:4857791	A/G	rs210475204	*COBL*	6.507	1.09	2.195 × 10^−9^
	4:9502576	C/T	rs445255852	*LRRD1*	6.653	1.24	7.130 × 10^−8^
	4:9875221	G/A	rs379759182	*RBM48*	4.342	0.89	9.806 × 10^−7^
	4:7208133	C/T	rs516634298	*ABCA13*	3.780	0.79	1.666 × 10^−6^
	6:37403795	T/C	rs109696064	*LCORL*	13.712	1.49	3.011 × 10^−20^
	6:37343379	G/T	rs109570900	*NCAPG*	11.838	1.43	9.919 × 10^−17^
	6:36630884	T/C	rs383697460	*PKD2*	15.613	2.83	3.458 × 10^−8^
	6:58146321	G/A	rs797342426	Hypothetical	14.673	2.68	4.520 × 10^−8^
	6:36880429	C/T	rs716537943	*IBSP*	7.977	1.52	1.490 × 10^−7^
	6:36028197	G/A	rs383620650	*FAM13A*	8.228	1.59	2.346 × 10^−7^
	6:37237698	T/C	rs210785796	*FAM184B*	−4.537	0.88	2.626 × 10^−7^
	14:19524263	T/C	rs449968016	*PRKDC*	16.417	1.27	4.337 × 10^−38^
	14:30393332	C/A	rs109953090	*DNAJC5B*	10.964	1.10	2.686 × 10^−23^
	14:30518533	T/C	rs381116984	*CRH*	10.866	1.10	5.889 × 10^−23^
	14:33878123	A/G	rs464130691	*NCOA2*	10.022	1.34	8.765 × 10^−14^
	14:16328530	A/G	rs471616366	Hypothetical	10.655	1.47	4.335 × 10^−13^
	14:16641005	C/T	rs211636635	*TBC1D31*	10.450	1.47	1.045 × 10^−12^
	14:16698021	G/T	rs208131933	*TBC1D31*	10.450	1.47	1.045 × 10^−12^
	14:19984551	A/G	rs109071668	*PPDPFL*	4.505	0.69	7.512 × 10^−11^
	14:9051405	A/T	rs209264955	*HHLA1*	5.423	0.84	8.386 × 10^−11^
	14:19372860	T/C	rs210839501	*SPIDR*	3.949	0.66	2.213 × 10^−9^
	14:21195722	G/T	rs380004533	*ST18*	−4.275	0.75	9.922 × 10^−9^
	14:26475692	A/G	rs381829093	*CHD7*	−4.136	0.73	1.314 × 10^−8^
	14:31303419	G/A	rs109820067	*CSPP1*	−3.879	0.70	2.738 × 10^−8^
	14:43787543	A/G	rs380389290	Hypothetical	5.905	1.07	3.557 × 10^−8^
	14:9111310	T/C	rs209285140	*OC90*	4.037	0.73	3.609 × 10^−8^
	14:9111217	A/G	rs210209375	*OC90*	4.036	0.73	3.659 × 10^−8^
	14:16590597	T/C	rs207540257	*FAM83A*	6.487	1.18	3.713 × 10^−8^
	14:16590776	T/C	rs380808409	*FAM83A*	6.487	1.18	3.713 × 10^−8^
	14:9111232	G/A	rs207841625	*OC90*	3.968	0.72	4.009 × 10^−8^
	14:16609068	T/A	rs210725961	*FAM83A*	6.441	1.18	4.470 × 10^−8^
	14:24756697	T/C	rs136157938	*SDCBP*	−3.559	0.66	5.573 × 10^−8^
	14:30837447	C/T	rs109134892	*MYBL1*	−3.230	0.67	1.567 × 10^−6^
BFT	2:107160304	G/T	rs109446852	*ZFAND2B*	0.799	0.11	1.840 × 10^−12^
	2:107114443	A/G	rs383795443	*SLC23A3*	0.768	0.12	2.549 × 10^−11^
EMA	14:19524263	T/C	rs449968016	*PRKDC*	1.644	0.30	4.617 × 10^−8^
MS	3:19028381	G/A	rs210416891	*MRPL9*	0.251	0.05	1.408 × 10^−6^
	19:7310638	G/C	rs799291287	*ANKFN1*	−0.234	0.05	9.683 × 10^−7^
	22:11849704	G/C	rs799031002	*ACVR2B*	0.298	0.05	1.567 × 10^−8^

**Table 2 genes-14-01839-t002:** Contributions of the most significant SNP of each chromosome (BTA) for carcass traits.

Trait	CHROM	RefSNPID	Gene	%V(g)	%V(p)
CW	BTA4	rs210475204	*COBL*	1.634	0.005
	BTA6	rs109696064	*LCORL*	4.495	0.014
	BTA14	rs449968016	*PRKDC*	8.470	0.026
BFT	BTA2	rs109446852	*ZFAND2B*	2.075	0.006
EMA	BTA14	rs449968016	*PRKDC*	1.728	0.005
MS	BTA3	rs210416891	*MRPL9*	1.104	0.004
	BTA19	rs799291287	*ANKFN1*	0.566	0.002
	BTA22	rs799031002	*ACVR2B*	1.187	0.004

**Table 3 genes-14-01839-t003:** Effects of the most significant SNPs of each chromosome on protein structure in carcass traits.

Trait	Protein	HGVS	Predicted Classification ^1^	Damaging Score ^2^	Sensitivity	Specificity
CW	COBL	p.Arg660Gln	No damage	0.007	0.96	0.75
	LCORL	p.Glu485Lys	Unknown	Unknown	Unknown	Unknown
	PRKDC	p.Ala895Thr	Probably damaging	0.988	0.73	0.96
BFT	ZFAND2B	p.Leu117Arg	No damage	0.000	1.00	0.00
EMA	PRKDC	p.Ala895Thr	Probably damaging	0.988	0.73	0.96
MS	MRPL9	p.Thr226Ala	Probably damaging	0.998	0.27	0.99
	ANKFN1	p.Leu174Val	Probably damaging	0.961	0.78	0.95
	ACVR2B	p.Thr395Ser	Possibly damaging	0.734	0.85	0.92

^1^ Divided by the variant effect prediction score: No damage (benign), possibly damaging, and probably damaging. Probably damaging indicates that the variant is likely to affect protein function. ^2^ The probably damaging score ranges from 0.0 (tolerated) to 1.0 (deleterious). 0 to 0.15—variants with scores in this range are predicted to be no damage. 0.15 to 1.0—variants with scores in this range are possibly damaging. 0.85 to 1.0—variants with scores in this range are more confidently predicted to be damaging.

## Data Availability

No new data were created or analyzed in this study. Data sharing is not applicable to this article.

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
