# Peer review of "Identification of Missense Variants Affecting Carcass Traits for Hanwoo Precision Breeding"

_genes, 2023, doi:10.3390/genes14101839_

Round 1

Reviewer 1 Report

This study provides a comprehensive search for the impact of genomic variants on beef characteristics of Hanwoo cattle. The investigation considers the possible impact at the protein structure level.

Minor comments are:

Line comment

97   R2 is not defined.

117 If 0, 1, 2 is the number of one of the alleles, then so indicate.

124 Correct location of superscript 2

Author Response

Dear Reviewer.

Thanks for the review.

Some minor comments you made are as follows:
1. line 97 R2 is not defined.
Re: please check the attachment file (line 98).

2. line 117 If 0, 1, 2 is the number of one of the alleles, then so indicate.
Re: please check the attachment file (line 117).

3. line 124 Correct location of superscript 2
Re: please check the attachment file (line 124).

Reviewer 2 Report

The article received for review is well prepared, written in a coherent and cursive manner, with high quality results obtained thanks to the methods used. Some minor aspects that escaped the authors, probably in a hurry, refer to the writing of names in Latin with italics (for example, Bos taurus, row 45) or ending some sentences with a comma instead of a full stop (row 49). In my opinion, the title is formulated in a way that is difficult to understand and that seems to detract from the importance of the achievements rather than enhance them.

The article received for review is well prepared, written in a coherent and cursive manner, with high quality results obtained thanks to the methods used. Some minor aspects that escaped the authors, probably in a hurry, refer to the writing of names in Latin with italics (for example, Bos taurus, row 45) or ending some sentences with a comma instead of a full stop (row 49). In my opinion, the title is formulated in a way that is difficult to understand and that seems to detract from the importance of the achievements rather than enhance them.

Author Response

Dear Reviewer.

Thanks for the review.

Some minor comments you made are as follows:

1. refer to the writing of names in Latin with italics (Bos taurus, row 45)
Re: please check the attachment file (line 45).

2. ending some sentences with a comma instead of a full stop (row 49).
Re: please check the attachment file (line 49).

3. the title is formulated in a way that is difficult to understand and that seems to detract from the importance of the achievements rather than enhance them.
Re: We have changed the title of the paper to reflect your opinion. Please check the attachment file.
